# Basis Refinement Strategies for Linear Value Function Approximation in MDPs

**Gheorghe Comanici**
School of Computer Science
McGill University
Montreal, Canada
gcoman@cs.mcgill.ca

**Doina Precup**
School of Computer Science
McGill University
Montreal, Canada
dprecup@cs.mcgill.ca

**Prakash Panangaden**
School of Computer Science
McGill University
Montreal, Canada
prakash@cs.mcgill.ca

## Abstract

We provide a theoretical framework for analyzing basis function construction for linear value function approximation in Markov Decision Processes (MDPs). We show that important existing methods, such as Krylov bases and Bellman-error-based methods are a special case of the general framework we develop. We provide a general algorithmic framework for computing basis function refinements which "respect" the dynamics of the environment, and we derive approximation error bounds that apply for any algorithm respecting this general framework. We also show how, using ideas related to bisimulation metrics, one can translate basis refinement into a process of finding "prototypes" that are diverse enough to represent the given MDP.

## 1  Introduction

Finding optimal or close-to-optimal policies in large Markov Decision Processes (MDPs) requires the use of approximation. A very popular approach is to use linear function approximation over a set of features [Sutton and Barto, 1998, Szepesvari, 2010]. An important problem is that of determining automatically this set of features in such a way as to obtain a good approximation of the problem at hand. Many approaches have been explored, including adaptive discretizations [Bertsekas and Castanon, 1989, Munos and Moore, 2002], proto-value functions [Mahadevan, 2005], Bellman error basis functions (BEBFs) [Keller et al., 2006, Parr et al., 2008a], Fourier basis [Konidaris et al., 2011], feature dependency discovery [Geramifard et al., 2011] etc. While many of these approaches have nice theoretical guarantees when constructing features for *fixed* policy evaluation, this problem is significantly more difficult in the case of optimal control, where multiple policies have to be evaluated using the same representation.

We analyze this problem by introducing the concept of *basis refinement*, which can be used as a general framework that encompasses a large class of iterative algorithms for automatic feature extraction. The main idea is to start with a set of basis which are consistent with the reward function, i.e. which allow only states with similar immediate reward to be grouped together. One-step look-ahead is then used to find parts of the state space in which the current basis representation is inconsistent with the environment dynamics, and the basis functions are adjusted to fix this problem. The process continues iteratively. We show that BEBFs [Keller et al., 2006, Parr et al., 2008a] can be viewed as a special case of this iterative framework. These methods iteratively expand an existing set of basis functions in order to capture the residual Bellman error. The relationship between such features and augmented Krylov bases allows us to show that every additional feature in these sets is consistently refining intermediate bases. Based on similar arguments, it can be shown that other methods, such as those based on the concept of MDP homomorphisms [Ravindran and Barto, 2002], bisimulation metrics [Ferns et al., 2004], and partition refinement algorithms [Ruan et al., 2015], are also special cases of the framework. We provide approximation bounds for sequences of refinements, as

well as a basis convergence criterion, using mathematical tools rooted in bisimulation relations and metrics [Givan et al., 2003, Ferns et al., 2004].

A final contribution of this paper is a new approach for computing alternative representations based on a selection of *prototypes* that incorporate all the necessary information to approximate values over the entire state space. This is closely related to kernel-based approaches [Ormoneit and Sen, 2002, Jong and Stone, 2006, Barreto et al., 2011], but we do not assume that a metric over the state space is provided (which allows one to determine similarity between states). Instead, we use an iterative approach, in which prototypes are selected to properly distinguish dynamics according to the current basis functions, then a new metric is estimated, and the set of prototypes is refined again. This process relies on using *pseudometrics* which in the limit converge to bisimulation metrics.

## 2 Background and notation

We will use the framework of Markov Decision Processes, consisting of a finite state space $S$, a finite action space $A$, a transition function $P : (S \times A) \to \mathbb{P}(S)$[1], where $P(s,a)$ is a probability distribution over the state space $S$, a reward function[2] $R : (S \times A) \to \mathbb{R}$. For notational convenience, $P^a(s), R^a(s)$ will be used to denote $P(s,a)$ and $R(s,a)$, respectively. One of the main objectives of MDP solvers is to determine a good action choice, also known as a policy, from every state that the system would visit. A policy $\pi : S \to \mathbb{P}(A)$ determines the probability of choosing each action $a$ given the state $s$ (with $\sum_{a \in A} \pi(s)(a) = 1$). The value of a policy $\pi$ given a state $s_0$ is defined as

$$V^\pi(s_0) = \mathbb{E} \left[ \sum_{i=0}^\infty \gamma^i R^{a_i}(s_i) \mid s_{i+1} \sim P^{a_i}(s_i), \ a_i \sim \pi(s_i) \right].$$

Note that $V^\pi$ is a real valued function $[\![S \to \mathbb{R}]\!]$; the space of all such functions will be denoted by $\mathcal{F}_S$. We will also call such functions *features*. Let $R^\pi$ and $P^\pi$ denote the reward and transition probabilities corresponding to choosing actions according to $\pi$. Note that $R^\pi \in \mathcal{F}_S$ and $P^\pi \in [\![\mathcal{F}_S \to \mathcal{F}_S]\!]$, where[3] $R^\pi(s) = \mathbb{E}_{a \sim \pi(s)}[R^a(s)]$ and $P^\pi(f)(s) = \mathbb{E}_{a \sim \pi(s)} \left[ \mathbb{E}_{P^a(s)}[f] \right]$. Let $T^\pi \in [\![\mathcal{F}_S \to \mathcal{F}_S]\!]$ denote the Bellman operator: $T^\pi(f) = R^\pi + \gamma P^\pi(f)$. This operator is linear and $V^\pi$ is its fixed point, i.e. $T^\pi(V^\pi) = V^\pi$. Most algorithms for solving MDPs will either use the model $(R^\pi, P^\pi)$ to find $V^\pi$ (if this model is available and/or can be estimated efficiently), or they will estimate $V^\pi$ directly using samples of the model, $\{(s_i, a_i, r_i, s_{i+1})\}_{i=0}^\infty$. The value $V^*$ associated with the best policy $\pi^*$ is the fixed point of the *Bellman optimality operator* $T^*$ (*not* a linear operator), defined as: $T^*(f) = \max_{a \in A} \left( R^a + \gamma P^a(f) \right)$.

The main problem we address in this paper is that of finding alternative representations for a given MDP. In particular, we look for finite, linearly independent subsets $\Phi$ of $\mathcal{F}_S$. These are **bases** for subspaces that will be used to speed up the search for $V^\pi$, by limiting it to $span(\Phi)$. We say that a basis $B$ is a **partition** if there exists an equivalence relation $\sim$ on $S$ such that $B = \{\chi(C) \mid C \in S/\sim\}$, where $\chi$ is the characteristic function (i.e. $\chi(X)(x) = 1$ if $x \in X$ and 0 otherwise). Given any equivalence relation $\sim$, we will use the notation $\Delta(\sim)$ for the set of characteristic functions on the equivalence classes of $\sim$, i.e. $\Delta(\sim) = \{\chi(C) \mid C \in S/\sim\}$.[4]

Our goal will be to find subsets $\Phi \subset \mathcal{F}_S$ which allow a value function approximation with strong quality guarantees. More precisely, for any policy $\pi$ we would like to approximate $V^\pi$ with $V_\Phi^\pi = \sum_{i=1}^k w_i \phi_i$ for some choice of $w_i$'s, which amounts to finding the best candidate inside the space spanned by $\Phi = \{\phi_1, \phi_2, ..., \phi_k\}$. A sufficient condition for $V^\pi$ to be an element of $span(\Phi)$ (and therefore representable exactly using the chosen set of bases), is for $\Phi$ to span the reward function and be an invariant subspace of the transition function: $R^\pi \in span(\Phi)$ and $\forall f \in \Phi, \ P^\pi(f) \in span(\Phi)$. *Linear fixed point methods* like TD, LSTD, LSPE [Sutton, 1988, Bradtke and Barto, 1996, Yu and Bertsekas, 2006] can be used to find the *least squares fixed point approximation* $V_\Phi^\pi$ of $V^\pi$ for a representation $\Phi$; these constitute proper approximation schemes, as

one can determine the number of iterations required to achieve a desired approximation error. Given a representation $\Phi$, the approximate value function $V_\Phi^\pi$ is the fixed point of the operator $T_\Phi^\pi$, defined as: $T_\Phi^\pi f := \Pi_\Phi(R^\pi + \gamma P^\pi(f))$, where $\Pi_\Phi$ is the orthogonal projection operator on $\Phi$. Using the linearity of $\Pi_\Phi$, it directly follows that $T_\Phi^\pi(f) = \Pi_\Phi R^\pi + \gamma \Pi_\Phi P^\pi(f)$ and $V_\Phi^\pi$ is the fixed point of the Bellman operator over the transformed linear model $(R_\Phi^\pi, P_\Phi^\pi) := (\Pi_\Phi R^\pi, \Pi_\Phi P^\pi)$. For more details, see [Parr et al., 2008a,b].

The analysis tools that we will use to establish our results are based on probabilistic bisimulation and its quantitative analogues. Strong probabilistic bisimulation is a notion of behavioral equivalence between the states of a probabilistic system, due to [Larsen and Skou, 1991] and applied to MDPs with rewards by [Givan et al., 2003]. The metric analog is due to [Desharnais et al., 1999, 2004] and the extension of the metric to include rewards is due to [Ferns et al., 2004]. An equivalence relation $\sim$ is a *a bisimulation relation* on the state space $S$ if for every pair $(s, s') \in S \times S$, $s \sim s'$ if and only if $\forall a \in A, \forall C \in S/\sim$, $R^a(s) = R^a(s')$ and $P^a(s)(C) = P^a(s')(C)$ (we use here $P^a(s)(C)$ to denote the probability of transitioning into $C$, under transition $s, a$). A pseudo-metric is a *bisimulation metric* if there exists some bisimulation relation $\sim$ such that $\forall s, s', d(s, s') = 0 \iff s \sim s'$.

The bisimulation metrics described by [Ferns et al., 2004] are constructed using the Kantorovich metric for comparing two probability distributions. Given a ground metric $d$ over $S$, the Kantorovich metric over $\mathbb{P}(S)$ takes the largest difference in the expected value of Lipschitz-1 functions with respect to $d$: $\Omega(d) := \{f \in \mathcal{F}_S \mid \forall s, s', f(s) - f(s') \le d(s, s')\}$. The distance between two probabilities $\mu$ and $\nu$ is computed as: $\mathcal{K}(d) : (\mu, \nu) \mapsto \sup_{\varphi \in \Omega(d)} \mathbb{E}_\mu[\varphi] - \mathbb{E}_\nu[\varphi]$. For more details on the Kantorovich metric, see [Villani, 2003]. The following approximation scheme converges to a bisimulation metric (starting with $d_0 = 0$, the metric that associates 0 to all pairs):

$$d_{k+1}(s, s') = \mathcal{T}(d_k)(s, s') := \max_a \left( (1 - \gamma) \left| R^a(s) - R^a(s') \right| + \gamma \mathcal{K}(d_k) \left( P^a(s), P^a(s') \right) \right). \quad (1)$$

The operator $\mathcal{T}$ has a fixed point $d^*$, which is a bisimulation metric, and $d_k \to d^*$ as $k \to \infty$. [Ferns et al., 2004] provide bounds which allow one to assess the quality of general state aggregations using this metric. Given a relation $\sim$ and its corresponding partition $\Delta(\sim)$, one can define an MDP model over $\Delta(\sim)$ as: $\hat{R}^a = \Pi_{\Delta(\sim)} R^a$ and $\hat{P}^a = \Pi_{\Delta(\sim)} P^a, \forall a \in A$. The approximation error between the true MDP optimal value function $V^*$ and its approximation using this reduced MDP model, denoted by $V^*_{\Delta(\sim)}$, is bounded above by:

$$\left| V^*_{\Delta(\sim)}(s) - V^*(s) \right| \le \frac{1}{1 - \gamma} d^*_\sim(s) + \max_{s' \in S} \frac{\gamma}{(1 - \gamma)^2} d^*_\sim(s'). \quad (2)$$

where $d^*_\sim(s)$ is average distance from a state $s$ to its $\sim$-equivalence class, defined as an expectation over the uniform distribution $\mathcal{U}$: $d^*_\sim(s) = \mathbb{E}_{\hat{s} \sim \mathcal{U}}[d^*(s, \hat{s}) \mid s \sim \hat{s}]$. Similar bounds for representations that are not partitions can be found in [Comanici and Precup, 2011]. Note that these bounds are minimized by aggregating states which are "close" in terms of the bisimulation distance $d^*$.

## 3  Basis refinement

In this section we describe the proposed basis refinement framework, which relies on "detecting" and "fixing" inconsistencies in the dynamics induced by a given set of features. Intuitively, states are *dynamically consistent* with respect to a set of basis functions if transitions out of these states are evaluated the same way by the model $\{P^a \mid a \in A\}$. Inconsistencies are "fixed" by augmenting a basis with features that are able to distinguish inconsistent states, relative to the initial basis. We are now ready to formalize these ideas.

**Definition 3.1.** Given a subset $F \subset \mathcal{F}_S$, two states $s, s' \in S$ are **consistent with respect to** $F$, denoted $s \sim_F s'$, if $\forall f \in F, \forall a \in A, f(s) = f(s')$ and $\mathbb{E}_{P^a(s)}[f] = \mathbb{E}_{P^a(s')}[f]$.

**Definition 3.2.** Given two subspaces $F, G \subset \mathcal{F}_S$, $G$ **refines** $F$ in an MDP $M$, and write $F \ltimes G$, if $F \subseteq G$ and

$$\forall s, s' \in S, \ s \sim_F s' \iff [\forall g \in G, \ g(s) = g(s')].$$

Using the linearity of expectation, one can prove that, given two probability distributions $\mu, \nu$, and a finite subset $\Gamma \subset F$, if $span(\Gamma) = F$, then $[\forall f \in F, \ \mathbb{E}_\mu[f] = \mathbb{E}_\nu[f]] \iff [\forall b \in \Gamma, \ \mathbb{E}_\mu[b] = \mathbb{E}_\nu[b]]$. For the special case of Dirac distributions $\delta_s$ and $\delta_{s'}$, for which

$\mathbb{E}_{\delta_s}[f] = f(s)$, it also holds that $\left[\forall f \in F, \quad f(s) = f(s')\right] \iff \left[\forall b \in \Gamma, \quad b(s) = b(s')\right]$. Therefore, Def. 3.2 gives a relation between two subspaces, but the refinement conditions could be checked on any basis choice. It is the subspace itself rather than a particular basis that matters, i.e. $\Gamma \ltimes \Gamma'$ if $span(\Gamma) \ltimes span(\Gamma')$. To fix inconsistencies on a pair $(s, s')$, for which we can find $f \in \Gamma$ and $a \in A$ such that either $f(s) \neq f(s')$ or $\mathbb{E}_{P^a(s)}[f] \neq \mathbb{E}_{P^a(s')}[f]$, one should construct a new function $\varphi$ with $\varphi(s) \neq \varphi(s')$ and add it to $\Gamma'$. To guarantee that all inconsistencies have been addressed, if $\varphi(s) \neq \varphi(s')$ for some $\varphi \in \Gamma'$, $\Gamma$ must contain a feature $f$ such that, for some $a \in A$, either $f(s) \neq f(s')$ or $\mathbb{E}_{P^a(s)}[f] \neq \mathbb{E}_{P^a(s')}[f]$.

In Sec. 5 we present an algorithmic framework consisting of sequential improvement steps, in which a current basis $\Gamma$ is refined into a new one, $\Gamma'$, with $span(\Gamma) \ltimes span(\Gamma')$. Def 3.2 guarantees that following such strategies expands $span(\Gamma)$ and that the approximation error for any policy will be decreased as a result. We now discuss bounds that can be obtained based on these definitions.

## 3.1 Value function approximation results

One simple way to create a refinement is to add to $\Gamma$ a single element that would address all inconsistencies: a feature that is valued differently for every element of $\Delta(\sim_\Gamma)$. Given $\omega : \Delta(\sim_\Gamma) \to \mathbb{R}$, $\left[\forall b, b' \in \Delta(\sim_\Gamma), \ b \neq b' \Rightarrow \omega(b) \neq \omega(b')\right] \Rightarrow \Gamma \ltimes \Gamma \cup \left\{\sum_{b \in \Delta(\sim_\Gamma)} \omega(b)b\right\}$. On the other hand, such a construction provides no approximation guarantee for the optimal value function (unless we make additional assumptions on the problem - we will discuss this further in Section 3.2). Although it addresses inconsistencies in the dynamics over the set of features spanned by $\Gamma$, it does not necessarily provide the representation power required to properly approximate the value of the optimal policy. The main theoretical result in this section provides conditions for describing refining sequences of bases, which are not necessarily accurate, but have approximation errors bounded by an exponentially decreasing function. These results are based on $\Delta(\sim_\Gamma)$, the largest basis refining subspace: any feature that is constant over equivalence classes of $\sim_\Gamma$ will be spanned by $\Delta(\sim)$, i.e. for any refinement $V \ltimes W$, $V \subseteq W \subseteq span(\Delta(\sim_V))$. These subsets are convenient as they can be analyzed using the bisimulation metric introduced in [Ferns et al., 2004].

**Lemma 3.1.** *The bisimulation operator in Eq. 1) is a contraction with constant $\gamma$. That is, for any metric $d$ over $S$, $\sup_{s, s' \in S} |\mathcal{T}(d)(s, s')| \leq \gamma \sup_{s, s' \in S} |d(s, s')|$.*

The proof relies on the Monge-Kantorovich duality (see [Villani, 2003]) to check that $\mathcal{T}$ satisfies sufficient conditions to be a contraction operator. An operator $Z$ is a contraction (with constant $\gamma < 1$) if $Z(x) \leq Z(x')$ whenever $x \leq x'$, and if $Z(x + c) = Z(x) + \gamma c$ for any constant $c \in \mathbb{R}$ [Blackwell, 1965]. One could easily check these conditions on the operator in Equation 1.

**Theorem 3.1.** *Let $\sim_0$ represent reward consistency, i.e. $s \sim_0 s' \iff \forall a \in A, R^a(s) = R^a(s')$, and $\Gamma_1 = \Delta(\sim_0)$. Additionally, assume $\{\Gamma_n\}_{n=1}^\infty$ is a sequence of bases such that for all $n \geq 1$, $\Gamma_n \ltimes \Gamma_{n+1}$ and $\Gamma_{n+1}$ is as large as the partition corresponding to consistency over $\Gamma_n$, i.e. $|\Gamma_{n+1}| = |S/\sim_{\Gamma_n}|$. If $V_{\Gamma_n}^*$ is the optimal value function computed with respect to representation $\Gamma_n$, then $\left\|V_{\Gamma_n}^* - V^*\right\|_\infty \leq \gamma^{n+1} \sup_{s, s', a} |R^a(s) - R^a(s')|/(1-\gamma)^2$.*

*Proof.* We will use the bisimulation metric defined in Eq. 1 and Eq. 2 applied to the special case of reduced models over bases $\{\Gamma_n\}_{n=1}^\infty$.

First, note that Monge-Kantorovich duality is crucial in this proof. It basically states that the Kantorovich metric is a solution to the Monge-Kantorovich problem, when its cost function is equal to the base metric for the Kantorovich metric. Specifically, for two measures $\mu$ and $\nu$, and a cost function $f \in [S \times S \to \mathbb{R}]$, the Monge-Kantorovich problem computes:

$$\mathcal{J}(f)(\mu, \nu) = \inf\{\mathbb{E}_\xi[f(x, y)] \mid \xi \in \mathbb{P}(S \times S) \text{ s.t. } \mu, \nu \text{ are the marginals corresponding to } x \text{ and } y\}$$

The set of measures $\xi$ with marginals $\mu$ and $\nu$ is also known as the set of couplings of $\mu$ and $\nu$. For any metric $d$ over $S$, $\mathcal{J}(d)(\mu, \nu) = \mathcal{K}(d)(\mu, \nu)$ (for proof, see [Villani, 2003]).

Next, we describe a relation between the metric $\mathcal{T}^n(0)$ and $\Gamma_n$. Since $|\Gamma_{n+1}| = |S/\sim_{\Gamma_n}| = |\Delta(\sim_{\Gamma_n})|$ and $\Gamma_{n+1} \subseteq span(\Delta(\sim_{\Gamma_n}))$, it must be the case that $span(\Gamma_{n+1}) = span(\Delta(\sim_{\Gamma_n}))$. It is not hard to see that for the special case of partitions, a refinement can be determined based on transitions into equivalence classes. Given

two equivalence relations $\sim_1$ and $\sim_2$, the refinement $\Delta(\sim_1) \ltimes \Delta(\sim_2)$ holds if and only if $s \sim_2 s' \Rightarrow s \sim_1 s'$ and $s \sim_2 s' \Rightarrow \left[\forall a \in A, \forall C \in S/\sim_1 \quad P^a(s)(C) = P^a(s')(C)\right]$. In particular, $\forall s, s'$ with $s \sim_{\Gamma_{n+1}} s'$, and $\forall C \in S/\sim_{\Gamma_n}, P^a(s)(C) = P^a(s')(C)$. This equality is crucial in defining the following coupling for $\mathcal{J}(f)(P^a(s), P^a(s'))$: let $\xi_C \in \mathbb{P}(S \times S)$ be any coupling of $P^a(s)|_C$ and $P^a(s')|_C$, the restrictions of $P^a(s)$ and $P^a(s')$ to $C$; the latter is possible as the two distributions are equal. Next, define the coupling $\xi$ of $\mu$ and $\nu$ as $\xi = \sum_{C \in S/\sim_{\Gamma_n}} \xi_C$. For any cost function $f$, if $s \sim_{\Gamma_{n+1}} s'$, then $\mathcal{J}(f)(P^a(s), P^a(s')) \leq \sum_{C \in S/\sim_{\Gamma_n}} E_{\xi_C}[f]$.

Using an inductive argument, we will now show that $\forall n, \ s \sim_{\Gamma_n} s' \Rightarrow \mathcal{T}^n(0)(s, s') = 0$. The base case is clear from the definition: $s \sim_0 s' \Rightarrow \mathcal{T}(0)(s, s') = 0$. Now, assume the former holds for $n$; that is, $\forall C \in S/\sim_{\Gamma_n}, \ \forall s, s' \in C, \mathcal{T}^n(0)(s, s') = 0$. But $\xi_C$ is zero everywhere except on the set $C \times C$, so $E_{\xi_C}[\mathcal{T}^n(0)] = 0$. Combining the last two results, we get the following upper bound:

$$s \sim_{\Gamma_{n+1}} s' \Rightarrow \mathcal{J}(\mathcal{T}^n(0))(P^a(s), P^a(s')) \leq \sum_{C \in S/\sim_{\Gamma_n}} E_{\xi_C}[\mathcal{T}^n(0)] = 0.$$

Since $\mathcal{T}^n(0)$ is a metric, it also holds that $\mathcal{J}(\mathcal{T}^n(0))(P^a(s), P^a(s')) \geq 0$. Moreover, as $s$ and $s'$ are consistent over $\Gamma_n \supseteq \Delta(\sim_0)$, this pair of states agree on the reward function. Therefore, $\mathcal{T}^{n+1}(0)(s, s') = \max_a((1 - \gamma)|R^a(s) - R^a(s')| + \gamma \mathcal{J}(\mathcal{T}^n(0))(P^a(s), P^a(s'))) = 0$.

Finally, for any $b \in \Delta(\sim_{\Gamma_n})$ and $s \in S$ with $b(s) = 1$, and any other state $\hat{s}$ with $b(\hat{s}) = 1$, it must be the case that $s \sim_{\Gamma_n} \hat{s}$ and $\mathcal{T}^n(0)(s, \hat{s}) = 0$. Therefore,

$$\mathop{\mathbb{E}}_{\hat{s} \sim \mathcal{U}} [d^*(s, \hat{s}) \mid s \sim_{\Gamma_n} \hat{s}] = \mathop{\mathbb{E}}_{\hat{s} \sim \mathcal{U}} [d^*(s, \hat{s}) - \mathcal{T}^n(0)(s, \hat{s}) \mid s \sim_{\Gamma_n} \hat{s}] \leq ||d^* - \mathcal{T}^n(0)||_\infty. \quad (3)$$

As $span(\Gamma_n) = span(\Delta(\sim_n))$, $V^*_{\Gamma_n}$ is the optimal value function for the MDP model over $\Delta(\sim_n)$. Based on (2) and (3), we can conclude that

$$\left|\left|V^*_{\Gamma_n} - V^*\right|\right|_\infty \leq \gamma ||d^* - \mathcal{T}^n(0)||_\infty / (1 - \gamma)^2. \quad (4)$$

But we already know from Lemma 3.1 that $d^*$ (defined in Eq. 1) is the fixed point of a contraction operator with constant $\gamma$. As $\mathcal{J}(0)(\mu, \nu) = 0$, the following holds for all $n \geq 1$

$$||d^* - \mathcal{T}^n(0)||_\infty \leq \gamma^n ||\mathcal{T}(0) - 0||_\infty / (1 - \gamma) \leq \gamma^n \sup_{s, s', a} |R^a(s) - R^a(s')|. \quad (5)$$

The final result is easily obtained by putting together Equations 4 and 5. $\qquad\square$

The result of the theorem provides a strategy for constructing refining sequences with strong approximation guarantees. Still, it might be inconvenient to generate refinements as large as $S/\sim_{\Gamma_n}$, as this might be over-complete; although faithful to the assumptions of the theorem, it might generate features that distinguish states that are not often visited, or pairs of states which are only slightly different. To address this issue, we provide a variation on the concept of refinement that can be used to derive more flexible refining algorithms: refinements that concentrate on local properties.

**Definition 3.3.** Given a subset $F \subset \mathcal{F}_S$, and a subset $\zeta \subset S$, two states $s, s' \in S$ are **consistent on $\zeta$ with respect to $F$**, denoted $s \sim_{F,\zeta} s'$, if $\forall f \in F, \forall a \in A, \ f(s) = f(s')$ and $\forall \hat{s} \in \zeta, \ \mathbb{E}_{P^a(\hat{s})}[f] = \mathbb{E}_{P^a(s)}[f] \iff \mathbb{E}_{P^a(\hat{s})}[f] = \mathbb{E}_{P^a(s')}[f]$.

**Definition 3.4.** Given two subspaces $F, G \subset \mathcal{F}_S$, $G$ **refines $F$ locally with respect to $\zeta$**, denoted $F \ltimes_\zeta G$, if $F \subseteq G$ and $\forall s, s' \in S, \ s \sim_{F,\zeta} s' \iff [\forall g \in G, \ g(s) = g(s')]$.

Definition 3.2 is the special case of Definition 3.4 corresponding to a refinement with respect to the whole state space $S$, i.e. $F \ltimes G \equiv F \ltimes_S G$. When the subset $\zeta$ is not important, we will use the notation $V \ltimes_\circ W$ to say that $W$ refines $V$ locally with respect to some subset of $S$. The result below states that even if one provides local refinements $\ltimes_\circ$, one will eventually generate a pair of subspaces which are related through a global refinement property $\ltimes$.

**Proposition 3.1.** Let $\{\Gamma_i\}_{i=0}^n$ be a set of bases over $S$ with $\Gamma_{i-1} \ltimes_{\zeta_i} \Gamma_i, i = 1, ..., n$, for some $\{\zeta_i\}_{i=1}^n$. Assume that $\Gamma_n$ is the maximal refinement (i.e. $|\Gamma_n| = |S/\sim_{\Gamma_{n-1}, \zeta_n}|$). Let $\eta = \cup_i \zeta_i$. Then $\Delta(\sim_{\Gamma_0, \eta}) \subseteq span(\Gamma_n)$.

*Proof.* Assume $s \sim_{\Gamma_{n-1}, \zeta_n} s'$. We will check below all conditions necessary to conclude that $s \sim_{\Gamma_0, \eta} s'$. First, let $f \in \Gamma_0$. It is immediate from the definition of local refinements that $\forall j \leq n - 1, \Gamma_j \subseteq \Gamma_{n-1}$, so that $s \sim_{\Gamma_0, \zeta_n} s'$. It follows that $\forall f \in \Gamma_0, \ f(s) = f(s')$.

Next, fix $f \in \Gamma_0$, $a \in A$ and $\hat{s} \in \eta$. If $\hat{s} \in \zeta_n$, then $\mathbb{E}_{P^a(\hat{s})}[f] = \mathbb{E}_{P^a(s)}[f] \iff \mathbb{E}_{P^a(\hat{s})}[f] = \mathbb{E}_{P^a(s')}[f]$, by the assumption above on the pair $s, s'$. Otherwise, $\exists j < n$ such that $\hat{s} \in \zeta_j$ and $\Gamma_{j-1} \ltimes_{\zeta_j} \Gamma_j$. But we already know that $\forall f \in \Gamma_j$, $f(s) = f(s')$, as $\Gamma_j \subseteq \Gamma_{n-1}$. We can use this result in the definition of local refinement $\Gamma_{j-1} \ltimes_{\zeta_j} \Gamma_j$ to conclude that $s \sim_{\Gamma_{j-1}, \zeta_j} s'$. Moreover, as $\hat{s} \in \zeta_j$, $f \in \Gamma_0 \subseteq \Gamma_{j-1}$, $\mathbb{E}_{P^a(\hat{s})}[f] = \mathbb{E}_{P^a(s)}[f] \iff \mathbb{E}_{P^a(\hat{s})}[f] = \mathbb{E}_{P^a(s')}[f]$. This completes the definition of consistency on $\eta$, and it becomes clear that $s \sim_{\Gamma_{n-1}, \zeta_n} s' \Rightarrow s \sim_{\Gamma_0, \eta} s'$, or $\Delta(\sim_{\Gamma_0, \eta}) \subseteq span(\Delta(\sim_{\Gamma_{n-1}, \zeta_n}))$.

Finally, both $\Gamma_n$ and $\Delta(\sim_{\Gamma_{n-1}, \eta})$ are bases of the same size, and both refine $\Gamma_{n-1}$. It must be that $span(\Gamma_n) = span(\Delta(\sim_{\Gamma_{n-1}, \zeta_n})) \supseteq \Delta(\sim_{\Gamma_0, \eta})$. $\qquad\square$

### 3.2  Examples of basis refinement for feature extraction

The concept of basis refinement is not only applicable to the feature extraction methods we will present later, but to methods that have been studied in the past. In particular, *methods based on Bellman error basis functions, state aggregation strategies, and spectral analysis using bisimulation metrics are all special cases of basis refinement*. We briefly describe the refinement property for the first two cases, and, in the next section, we elaborate on the connection between refinement and bisimulation metrics to provide a new condition for convergence to self-refining bases.

***Krylov bases:*** Consider the uncontrolled (policy evaluation) case, in which one would like to find a set of features that is suited to evaluating a single policy of interest. A common approach to automatic feature generation in this context computes Bellman error basis functions (BEBFs), which have been shown to generate a sequence of representations known as *Krylov bases*. Given a policy $\pi$, a Krylov basis $\Phi_n$ of size $n$ is built using the model $(R^\pi, P^\pi)$ (defined in Section 2 as elements of $\mathcal{F}_S$ and $[\![\mathcal{F}_S \to \mathcal{F}_S]\!]$, respectively): $\Phi_n = span\{R^\pi, P^\pi R^\pi, (P^\pi)^2 R^\pi, ..., (P^\pi)^n R^\pi\}$. It is not hard to check that $\Phi_n \ltimes \Phi_{n+1}$, where $\ltimes$ is the refinement relational property in Def 3.2. Since the initial feature $R^\pi \in \Delta(\sim_0)$, the result in Theorem 3.1 holds for the Krylov bases.

Under the assumption of a finite-state MDP (i.e. $|S| < \infty$), $\Gamma_\chi := \{\chi(\{s\}) \mid s \in S\}$ is a basis for $\mathcal{F}_S$, therefore this set of features is finite dimensional. It follows that one can find $N \le |S|$ such that one of the Krylov bases is *a self-refinement*, i.e. $\Phi_N \ltimes \Phi_N$. This would by no means be the only self-refining basis. In fact this property holds for the basis of characteristic functions, $\Gamma_\chi \ltimes \Gamma_\chi$. The purpose our framework is to determine other self-refining bases which are suited for function approximation methods in the context of controlled systems.

***State aggregation:*** One popular strategy used for solving MDPs is that of computing state aggregation maps. Instead of working with alternative subspaces, these methods first compute equivalence relations on the state space. An aggregate/collapsed model is then derived, and the solution to this model is translated to one for the original problem: the resulting policy provides the same action choice for states that have originally been related. Given any equivalence relation $\sim$ on $S$, a state aggregation map is a function from $S$ to any set $X$, $\rho : S \to X$, such that $\forall s, s'$, $\rho(s') = \rho(s) \iff s \sim s'$. In order to obtain a significant computational gain, one would like to work with aggregation maps $\rho$ that reduce the size of the space for which one looks to provide action choices, i.e. $|X| \ll |S|$. As discussed in Section 3.1, one could work with features that are defined on an aggregate state space instead of the original state space. That is, instead of computing a set of state features $\Gamma \subset \mathcal{F}_S$, we could work instead with an aggregation map $\rho : S \to X$ and a set of features over $X$, $\hat{\Gamma} \subset \mathcal{F}_X$. If $\sim$ is the relation such that $s \sim s' \iff \rho(s) = \rho(s')$, then $\forall \varphi \in \hat{\Gamma}$, $\varphi \circ \rho \in span(\Delta(\sim))$.

## 4  Using bisimulation metrics for convergence of bases

In Section 3.2 we provide two examples of self-refining subspaces: the Krylov bases and the characteristic functions on single states. The latter is the largest and sparsest basis; it spans the entire state space and the features share no information. The former is potentially smaller and it spans the value of the fixed policy for which it was designed. In this section we will present a third self-refining construction, which is designed to capture bisimulation properties. Based on the results presented in Section 3.1, it can be shown that given a bisimulation relation $\sim$, the partition it generates is self-refining, i.e. $\Delta(\sim) \ltimes \Delta(\sim)$.

Desirable self-refining bases might be be computationally demanding and/or too complex to use or represent. We propose iterative schemes which ultimately provide a self-refining result - albeit we would have the flexibility of stopping the iterative process before reaching the final result. At the same time, we need a criterion to describe convergence of sequences of bases. That is, we would want to know *how close* an iterative process is to obtaining a self-refining basis. Inspired by the fixed point theory used to study bisimulation metrics [Desharnais et al., 1999], instead of using a metric over the set of all bases to characterize convergence of such sequences, we will use corresponding metrics over the original state space. This choice is better suited for generalizing previously existing methods that compare pairs of states for bisimilarity through their associated reward models and expected realizations of features over the next state distribution model associated with these states. We will study metric construction strategies based on a map $\mathcal{D}$, defined below, which takes an element of the powerset $\mathcal{P}(\mathcal{F}_S)$ of $\mathcal{F}_S$ and returns an element of all pseudo-metrics $\mathscr{M}(S)$ over $S$.

$$\mathcal{D}(\Gamma) : (s, s') \mapsto \max_a \left[ (1-\gamma) \left| R^a(s) - R^a(s') \right| + \gamma \sup_{\varphi \in \Gamma} \left| \mathbb{E}_{P^a(s)}[\varphi] - \mathbb{E}_{P^a(s')}[\varphi] \right| \right] \quad (6)$$

$\Gamma$ is a set of features whose expectation over next-state distributions should be matched. It is not hard to see that bases $\Gamma$ for which $\mathcal{D}(\Gamma)$ is a bisimulation metric are by definition self-refining. For example, consider the largest bisimulation relation $\sim$ on a given MDP. It is not hard to see that $\mathcal{D}(\Delta(\sim))$ is a bisimulation. A more elaborate example involves the set $\Omega(d)$ of Lipschitz-1 continuous functions on $[\![(S, d) \to (\mathbb{R}, L_1)]\!]$ (recall definition and computation details from Section 2). Define $d^*$ to be the fixed point of the operator $T : d \mapsto \mathcal{D}(\Omega(d))$, i.e. $d^* = \sup_{n \in \mathbb{N}} T^n(0)$. $d^*$ has the same property as the bisimulation metric defined in Equation 1. Moreover, given any bisimulation metric $d$, $\mathcal{D}(\Omega(d))$ is a bisimulation metric.

**Definition 4.1.** We say a sequence $\{\Gamma_n\}_{n=1}^\infty$ is a *a bisimulation sequence of bases* if $\mathcal{D}(\Gamma_n)$ converges uniformly from below to a bisimulation metric. If one has the a sequence of refining bases with $\Gamma_n \ltimes \Gamma_{n+1}, \forall n$, then $\{\mathcal{D}(\Gamma_n)\}_{n=1}^\infty$ is an increasing sequence, but not necessarily a bisimulation sequence.

A bisimulation sequence of bases provide an approximation scheme for bases that satisfy two important properties studied in the past: self-refinement and bisimilarity. One could show that the approximation schemes presented in [Ferns et al., 2004], [Comanici and Precup, 2011], and [Ruan et al., 2015] are all examples of bisimulation sequences. We will present in the next section a framework that generalizes all these examples, but which can be easily extended to a broader set of approximation schemes that incorporate both refining and bisimulation principles.

## 5 Prototype based refinements

In this section we propose a strategy that iteratively builds sequences of refineing sets of features, based on the concepts described in the previous sections. This generates layered sets of features, where the $n^{\text{th}}$ layer in the construction will be dependent only on the $(n-1)^{\text{th}}$ layer. Additionally, each feature will be associated with a reward-transition ***prototype***: elements of $\mathcal{Q} := [\![A \to (\mathbb{R} \times \mathbb{P}(S))]\!]$, associating to each action a reward and a next-state probability distribution. Prototypes can be viewed as "abstract" or representative states, such as used in KBRL methods [Ormoneit and Sen, 2002]. In the layered structure, the similarity between prototypes at the $n^{\text{th}}$ layer is based on a measure of consistency with respect to features at the $(n-1)^{\text{th}}$ layer. The same measure of similarity is used to determine whether the entire state space is "covered" by the set of prototypes/features chosen for the $n^{\text{th}}$ layer. We say that a space is *covered* if every state of the space is close to at least one prototype generated by the construction, with respect to a predefined measure of similarity. This measure is designed to make sure that consecutive layers represent refining sets of features. Note that for any given MDP, the state space $S$ is embedded into $\mathcal{Q}$ (i.e. $S \subset \mathcal{Q}$), as $(R^a(s), P^a(s)) \in \mathcal{Q}$ for every state $s \in S$. Additionally, The metric generator $\mathcal{D}$, as defined in Equation 6, can be generalized to a map from $\mathcal{P}(\mathcal{F}_S)$ to $\mathscr{M}(\mathcal{Q})$.

The algorithmic strategy will look for a sequence $\{J_n, \iota_n\}_{n=1}^\infty$, where $J_n \subset \mathcal{Q}$ is a set of covering prototypes, and $\iota_n : J_n \to \mathcal{F}_S$ is a function that associates a feature to every prototype in $J_n$. Starting with $J_0 = \emptyset$ and $\Gamma_0 = \emptyset$, the strategy needs to find, at step $n > 0$, a cover $\hat{J}_n$ for $S$, based on the distance metric $\mathcal{D}(\Gamma_{n-1})$. That is, it has to guarantee that $\forall s \in S, \exists \kappa \in \hat{J}_n$ with $\mathcal{D}(\Gamma_{n-1})(s, \kappa) = 0$. With $J_n = \hat{J}_n \cup J_{n-1}$ and using a strictly decreasing function $\tau : \mathbb{R}_{\geq 0} \to \mathbb{R}$ (e.g. the energy-based Gibbs measure $\tau(x) = \exp(-\beta x)$ for some $\beta > 0$), the framework constructs $\iota_n : J_n \to \mathcal{F}_S$, a map that associates prototypes to features as $\iota_n(\kappa)(s) = \tau(\mathcal{D}(\Gamma_{n-1})(\kappa, s))$.

---
**Algorithm 1** Prototype refinement
---
1: $J_0 = \emptyset$ and $\Gamma_0 = \emptyset$
2: **for** $n = 1$ to $\infty$ **do**
3:     choose a representative subset $\zeta_n \subset S$ and a cover approximation error $\epsilon_n \geq 0$
4:     find an $\epsilon_n$-cover $\hat{J}_n$ for $\zeta_n$
5:     define $J_n = \hat{J}_n \cup J_{n-1}$
6:     choose a strictly decreasing function $\tau : \mathbb{R}_{\geq 0} \to \mathbb{R}$
7:     define $\iota_n(\kappa) = \begin{cases} s \mapsto \tau(\mathcal{D}(\Gamma_{n-1})(\kappa, s)) & \text{if } \exists \hat{s} \in \zeta_n, \text{ such that } \mathcal{D}(\Gamma_{n-1})(\kappa, \hat{s}) \leq \epsilon_n \\ \iota_{n-1}(\kappa) & \text{otherwise} \end{cases}$
8:     define $\Gamma_n = \{\iota_n(\kappa) \mid \kappa \in J_n\}$ (note that $\Gamma_n$ is a local refinement, $\Gamma_{n-1} \ltimes_{\zeta_n} \Gamma_n$)
---

It is not hard to see that the refinement property holds at every step, i.e. $\Gamma_n \ltimes \Gamma_{n+1}$. First, every equivalence class of $\sim_{\Gamma_n}$ is represented by some prototype in $J_n$. Second, $\iota_n$ is purposely defined to make sure that a distinction is made between each prototype in $J_{n+1}$. Moreover, $\{\Gamma_n\}_{n=1}^{\infty}$ is a bisimulation sequence of bases, as the metric generator $\mathcal{D}$ is the main tool used in "covering" the state space with the set of prototypes $J_n$. Two states will be represented by the same prototype (i.e. they will be equivalent with respect to $\sim_{\Gamma_n}$) if and only if the distance between their corresponding reward-transition models is 0.

Algorithm 1 provides pseudo-code for the framework described in this section. Note that it also contains two additional modifications, used to illustrate the flexibility of this feature extraction process. Through the first modification, one could use the intermediate results at time step $n$ to determine a subset $\zeta_n \subset S$ of states which are likely to have a model with significantly distinct dynamics over $\Gamma_{n-1}$. As such, the prototypes $\hat{J}_{n-1}$ can be specialized to cover only the significant subset $\zeta_n$. Moreover Theorem 3.1 guarantees that if every state in $S$ is picked in $\zeta_n$ infinitely often, as $n \to \infty$, then the approximation power of the final result is not be compromised. The second modification is based on using the values in the metric $\mathcal{D}(\Gamma_{n-1})$ for more than just choosing feature activations: one could set at every step constants $\epsilon_n \geq 0$ and then find $J_n$ such that $\zeta_n$ is covered using $\epsilon_n$-balls, i.e. for every state in $\zeta_n$, there exists a prototype $\kappa \in J_n$ with $\mathcal{D}(\Gamma_{n-1})(\kappa, s) \leq \epsilon_n$. One can easily show that the refinement property can be maintained using the modified defition of $\iota_n$ described in Algorithm 1.

## 6   Discussion

We proposed a general framework for basis refinement for linear function approximation. The theoretical results show that *any* algorithmic scheme of this type satisfies strong bounds on the quality of the value function that can be obtained. In other words, this approach provides a "blueprint" for designing algorithms with good approximation guarantees. As discussed, some existing value function construction schemes fall into this category (such as state aggregation refinement, for example). Other methods, like BEBFs, can be interpreted in this way in the case of policy evaluation; however, the "traditional" BEBF approach in the case of control does not exactly fit this framework. However, we suspect that it could be adapted to exactly follow this blueprint (something we leave for future work).

We provided ideas for a new algorithmic approach to this problem, which would provide strong guarantees while being significantly cheaper than other existing methods with similar bounds (which rely on bisimulation metrics). We plan to experiment with this approach in the future. The focus of this paper was to establish the theoretical underpinnings of the algorithm. The algorithm structure we propose is close in spirit to [Barreto et al., 2011], which selects prototype states in order to represent well the dynamics of the system by means of stochastic factorization. However, their approach assumes a given metric which measures state similarity, and selects representative states using $k$-means clustering based on this metric. Instead, we iterate between computing the metric and choosing prototypes. We believe that the theory presented in this paper opens up the possibility of further development of algorithms for constructive function approximation that have quality guarantees in the control case, and which can be effective also in practice.

## Footnotes

[1]We will use $\mathbb{P}(X)$ to denote the set of probability distributions on a given set $X$.

[2]For simplicity, we assume WLOG that the reward is deterministic and independent of the state into which the system arrives.

[3]We will use $\mathbb{E}_\mu[f] = \sum_x f(x)\mu(x)$ to mean the expectation of a function $f$ wrt distribution $\mu$. If the function $f$ is multivariate, we will use $\mathbb{E}_{x \sim \mu}[f(x,y)] = \sum_x f(x,y)\mu(x)$ to denote expectation of $f$ when $y$ is fixed.

[4]The equivalence class of an element $s \in S$ is $\{s' \in S \mid s \sim s'\}$. $S/\sim$ is used for the quotient set of all equivalence classes of $\sim$.

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
