[Reviews · NeurIPS 2015]

Submitted by Assigned_Reviewer_1

Summary ========

The paper studies the problem of feature extraction/refinement for function approximation in reinforcement learning. In particular, the authors focus on linear approximation and rely on tools coming from bisimulation literature to describe the properties that a basis refinement process should have in order to guarantee a monotonically improving approximation of the optimal value function. In fact, the authors show that expanding a basis by using novel features that "solve" the inconsistencies of the current basis, it is possible to achieve an approximation of the optimal value function whose error decreases exponentially as gamma^n.

Review ========

The problem of desining suitable features/basis for RL problems is more delicate and less trivial that in other machine learning problems. As a result, developing methods for the automatic expansion of a given basis is a very interesting problem. This problem is not new and, as reviewed in the paper, it has been studied using different tools (more or less heuristic) where features are added incrementally so as to progressively reduce the approximation error. The paper provides a more comprehensive view of the problem of basis refinement and a nice theoretical framing of the problem. As such, the paper faces some shortcomings:

1- In the end, most of the paper is concerned with formalizing the problem and providing general theoretical guarantees for basis refinement. Only the very last part introduces an actual algorithm, but the algorithm details and the easiness of implementation remain relatively sketched. 2- There are already a few methods in the literature that fit into the proposed framework and have already been studied in theory and/or empirically. In this sense, the paper falls short in providing a detailed comparison between the methods and it is not clear to which extent the proposed "prototype-based" method could be a valid alternative to existing methods. 3- Methods such as those based on the Bellman-residual minimization are known to be tricky to implement, since expanding the current basis adding the current Bellman error as a new function is not trivial. In fact, "estimating" the Bellman residual may be quite difficult and it requires a sort of approximation step itself. This potential algorithmic issues are only rapidly treated in the paper.

Despite these limitations, I think the paper is a valuable step forward for a more formal and complete understanding of the problem of feature expansion in RL. The theory developed is not trivial and it provides a more solid grounding of some of the existing methods in the literature.
Summary: The paper introduces a general formalism to describe the problem of feature extraction/refinement for function approximation in RL. While the framework is still at a preliminary level, it unifies different existing approaches and it seems promising for the development of future methods.

Submitted by Assigned_Reviewer_2

This paper deals with basis construction for VFA, based on the concepts of equivalence relations and bisimulation metrics. The authors define a specific equivalence relation (consistency resp. to a subspace) and a concept of refinement and show (Th. 3.1) that starting from a reward equivalence relation, a sequence of refined bases (such that the cardinality of the newly generated basis is equal to the cardinality of the quotient set corresponding to the equivalence relation based on the span of the previous basis) will provide a (strong) guarantee regarding the quality of the optimal value function associated to this base. Then, they relax the introduced equivalence relation to take into account locality, and show a property on the link between local and global refinements. Next, they discuss a convergence criterion for this refinement principle based on bissimulation (equivalence and metric). Lastly, they extend the proposed refinement strategy by alternating metric learning and base learning (through prototypes).

The following detailed comments follow the paper (they are not ordered by importance). * l.81: s_i,a_i -> s_i,r_i ? * l.86: it may be useful to define the concepts of equivalence relation, equivalence class and quotient set (as they are a core of the paper) * l.116: s1 sim s2 -> s sim s' * l.116: precise that Pa(s)(C) is the measure of the set C * l.122: a missing absolute value in the def. of Omega(d)? * l.132: what is V^*_Delta? The least-squares fixed-point approximation (for the subspace Delta(sim)) of the optimal policy pi_*? How can this be computed? * l.157: beta is not defined and the notation Gamma is used afterwards * l.161-165: it is not clear how the refining fix inconsistencies in the dynamics (eg, if f(s) neq f(s'), take varphi = f, the subspace does not change) * l.173: it should be said that sim Sigma is sim span(Sigma) (even if rather clear from the context) * Th.3.1: how could be constructed the refinement? (the authors state that sec. 5 provides an answer, but see latter comments) Given the condition on the size of the refinement, how will the sequence of subspaces grows? Isn't it possible that a few refinements could lead to the whole space F_S (in which case the result is trivial)? * l.209: what is the (pseudo)-metric 0? the metric that associates 0 to any pair of states? * l.215: the notation P^a(s)|C is not defined (consequently, the rest of the proof is obscure) * l.231: where does the inequality in Eq. 3 come from? what is the supremum norm for the norm (entrywise, induced, other)? * l.233: what is V^*_n? is it V^*_Gamma_n? * l.235: in Eq.4, gamma/(1-gamma)^2 -> 1/(1-gamma)^2 ? * Prop.3.1: how is initialized the sequence? Above all, what is Gamma_(0,eta)? As it is not defined, it is hard to understand the stated result. * Sec.3.2: it is hard to see how Krylov bases/state aggregation relate to the stated result (apart from the addressed problem, that is basis construction). Do they satisfy the assumptions of Th.3.1? * l.314: Section 3.1 -> Section 3.2 * l.330: P and M are not defined * l.389: what is a supremum over metrics? * Def.4.1: it is hard to see the implications of this def., especially how it relates to a convergence criterion for subspaces refinement * Sec.5: the proposed approach is not clear at all. Providing a pseudo-code might help (and maybe instantiating the general approach too; for example how to find a cover over S, covering with what?). * l.367: what S subset Q means? why assuming it? * l.374: hat(J)n is not defined * l.376: iota(kappa) should be iota(kappa)(s). By the way, the metric D(Gamma) takes too states as argument, how can it takes a prototype kappa as the first argument? * l.423: significantly chapter?
Summary: This paper provides an abstract and theoretical framework for basis function construction for linear value function approximation (VFA). The paper is often hard to follow (partly due to unspecified objects, mainly due to being not familiar with bisimulation), it would benefit from providing more details (using the possibility to have an appendix, for example). Consequently, checking the correctness of the proofs is difficult, and it is also hard to tell to what extent the provided theoretical results are strong. The proposed framework is also really abstract, and it is far from being clear how to instantiate it to obtain practical algorithms (if possible).

Submitted by Assigned_Reviewer_3

Summary: this paper introduces a general framework for basis refinement for linear function approximation.

Quality/Clarity : the contribution of the paper is not very clear (what is a "framework" ?). The paper would definitely gain in quality if the contribution would be mure underlined.

Originality: Difficult to assess, since the paper looks like more a discussion rather than a new method.

Significance: I guess the discussion proposed in the paper is of interest to the community.

Additional comment: The present reviewer did not go through the proofs.
Summary: An interesting paper related with the basis refinement problem when using linear function approximation in reinforcement learning.

Submitted by Assigned_Reviewer_4

Basis Refinement Strategies for linear value function approximation in MDPs ------------------------------------------------------------ This paper provides a general theoretical framework for automatic basis function construction for linear value function approximation. The framework is shown to be a generalized version of different well known techniques already existing in this area such as Bellman Error Basis Functions (BEBF). The analysis gives theoretical guarantees when constructing features not only for fixed policy evaluation, which prior algorithms had done, but also for optimal control. This paper also contributes the outline for a new approach for computing alternative representations that is related to previous kernel-based approaches.

This paper provides new insight into basis construction algorithms. Building off of previous theoretical work in the area of bisimulations, it provides a solid theoretical framework that explains previous algorithms in this area. The new framework provides strong theoretical guarantees on the quality of the value function that can be obtained. The framework also provides guidelines on ways to develop new algorithms in this space with good approximation guarantees. The paper also gives some ideas on new ways to approach the problem of basis construction through refinements. The main point of the paper was the theoretical framework and the proofs, in which the paper did a good job. However, one of the main claims of contributions is the new algorithm ideas presented in section 5, and these ideas have not been tested. The authors do address this in the Discussion section, saying that they plan to experiment with these ideas in the future, but without the experiments, section 5 is definitely weak. Section 5 is also a bit confusing as the algorithm is outlined in prose instead of pseudo-code. In general, I found the paper extremely dense and hard to read.

There was not a lot of intuition provided.

It also does not seem that work such as the proto value functions or the Fourier basis can be captured in this framework.

There are a few minor typos and formatting errors. The title should be properly capitalized. The wrong word is used in the second paragraph of the discussions section. I'm assuming "...significantly chapter than other existing methods..." should read "...significantly cheaper than other existing methods...". Typos such as "Fourrier", "this type of features" etc should be fixed. While I didn't spot any errors in the mathematical notation, it would be a good idea to check over them one more time given the typos in the prose. Overall this seems like a good contribution to the area of RL, though the clarity and lack of experiments weakens it.
Summary: This is a good paper, though the lack of empirical results is a weakness.

Submitted by Assigned_Reviewer_5

The problem of automatically constructing a low dimensional approximation space for reinforcement learning has attracted considerable interest over the past few years. This paper develops some of these ideas further, bringing to bear some new tools based on bisimulation metrics, basis refinement etc. It provides some theoretical results on the quality of approximation produced by a learned basis.

On the whole, this is a rigorous and well written paper that seems to further our understanding of basis construction. However, the major flaw with this paper and one which precludes me from giving it a recommendation of acceptance, is that it contains no experimental results whatsoever. I think this is not sufficient for a NIPS paper, unless the quality of theoretical results is of such importance that the experiments are not necessary. I don't think this paper reaches that high standard when viewed purely as a theoretical paper.
Summary: The problem of automatically constructing a low dimensional approximation space for reinforcement learning has attracted considerable interest over the past few years. This paper develops some of these ideas further, bringing to bear some new tools based on bisimulation metrics, basis refinement etc. It provides some theoretical results on the quality of approximation produced by a learned basis.

Author Feedback
Author rebuttal: Thank you for the helpful reviews. We will incorporate the presentation suggestions in the paper.

The main issues pointed out by the reviewers are the lack of experiments and the lack of details and pseudo-code for the prototype-based refinement that we propose. We emphasize that the main contribution of the paper is the unifying framework of basis refinement: a framework that encompasses multiple automatic basis construction methods that have been published in the past. As such, published empirical results on these approaches already exist. It is true that we also provide a novel prototype-based algorithmic approach, in order to illustrate the flexibility of the proposed framework. It allows refining the state space in areas with "significant difference" in dynamics between states. A natural way to formalize this is by using bisimulation metrics, but other choices are possible. Hence, one can imagine multiple algorithms implementing this idea, which use different definitions and implementations for measuring "significant differences". We believe that a thorough presentation of these alternatives and empirical comparisons go beyond what can be explained clearly in the current paper, given space limitations. A simple illustration using one algorithmic choice could be included, but in our opinion it would weaken the presentation, since the theoretical results are already pretty dense. Therefore, we chose to omit experimental results from the paper, in order to explain the theoretical results clearly (since they are the main contribution). We can add a pseudocode description in Sec. 5 and a discussion of alternative implementations.

We omitted the discussion of certain methods for value function approximation from the paper due to lack of space, but we plan to address these in an expanded journal version. Briefly speaking, proto-value functions are designed to respect the dynamics of the environment but not the reward structure. Fourier bases are not specifically designed to respect either the reward, or the dynamics, as they can be defined based on the features only. We believe that a version of incremental feature dependency discovery that fits the prototype-based refinement can be designed, but this requires a lengthier discussion.

Reviewer 2 (R2) points out that the growth rate of the sequence of subspaces might be an issue. This is a very good point, and the reason for the approach outlined in Sec. 5, which is designed to control the growth. As discussed, there are many ways in which the refinement step can be implemented. The power of this theorem is that it applies to *any* refinement approach consistent with its conditions. Hence, it gives a proof "blueprint" for many different algorithms, including ones which may be developed in the future. Krylov bases satisfy its assumptions, yes (we will explain this more clearly).
Other clarifications for R2:
We will define concepts related to equivalences
The pseudo-metric 0 associates 0 to any pair of states
P^a(s)|C is 0 for any s outside C, and equal to P^a(s) for s in C.
122 needs an absolute value, yes.
V^*_\Delta is the optimal value function computed over the MDP with models \hat{R} and \hat{P} defined in line 131. This MDP is obtained by using the state aggregation defined by \Delta. We will clarify this.
beta is any set of bases (we will clarify).
We use _n as usual to denote iterative approximations (the approximation at the nth iteration)
In Pro. 3.1, the initialization is arbitrary, but usually, this would be a function with constant value for all states.
\cal{P} is the set of probability measures and \cal{M} is the set of (pseudo)metrics.
For any metric d over S, ||d||_\infty = \sup_{(s,s')\in S\times S} d(s,s')
We will add some further explanation of Def. 4.1
Q is just a set of all possible functions that produce a reward and a transition probability, so trivially each state has a unique associated such function (and can be viewed as this function)
\hat{J}_n is the new cover that needs to be found - we will clarify this.
Prototypes can be viewed as "abstract" or representative states (in the spirit of kernel methods, so we are overloading the notation slightly here. We will clarify this in the paper.